# DREAMLEARNING: DATA COMPRESSION ASSISTED MACHINE LEARNING

## ABSTRACT

Despite rapid advancements, machine learning, particularly deep learning, is hindered by the need for large amounts of labeled data to learn meaningful patterns without overfitting and immense demands for computation and storage, which motivate research into architectures that can achieve good performance with fewer resources. This paper introduces dreaMLearning, a novel framework that enables learning from compressed data without decompression, built upon Entropy-based Generalized Deduplication (EntroGeDe), an entropy-driven lossless compression method that consolidates information into a compact set of representative samples. dreaMLearning accommodates a wide range of data types, tasks, and model architectures. Extensive experiments on regression and classification tasks with tabular and image data demonstrate that dreaMLearning accelerates training by up to $10\times$, reduces peak memory usage of training data by $10\times$, and cuts storage by $37\%$, with a minimal impact on model performance. These advancements enhance diverse ML applications, including distributed and federated learning, and tinyML on resource-constrained edge devices, unlocking new possibilities for efficient and scalable learning.

## 1 INTRODUCTION

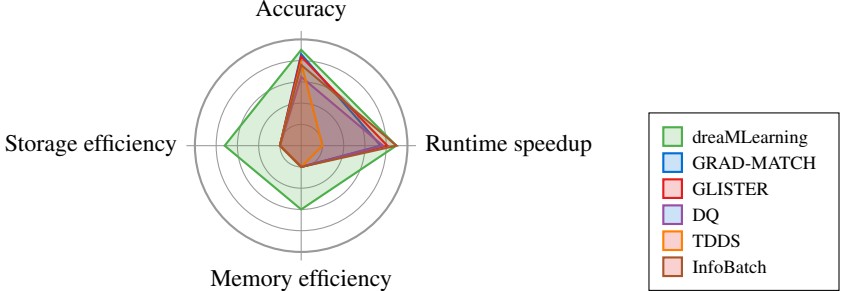

Figure 1: Performance benefits of dreaMLearning vs. coreset selection (CS) baselines, averaged over experimental results of CIFAR-10/100 and ImageNet-1k at a $10\%$ subset. dreaMLearning outperforms existing CS methods across all four key metrics, with pronounced gains in storage and memory efficiency enabled by direct training on compressed data (no full decompression or additional subset selection).

Scaling machine learning (ML) systems to handle larger datasets and more complex model architectures places increasing demands on computational system resources, e.g., storage, memory, and processing power Shen et al. (2024); Menghani (2023); Zhou et al. (2022); Nguyen et al. (2021). Training high-performance models

typically requires extensive and diverse datasets, but storing and accessing such data repeatedly is costly, especially in resource-constrained settings such as edge devices, federated learning.

Methods such as dataset distillation Wang et al. (2018); Cazenavette et al. (2022) and coreset selection Sener & Savarese (2018); Sinha et al. (2020); Coleman et al.; Toneva et al. (2019); Paul et al. (2021); Mirzasoleiman et al. (2020); Killamsetty et al. (2021a;b); Zhang et al. (2024); Qin et al. (2023) **trade-off accuracy for training speed-ups** by carrying out the training on a smaller number of samples, e.g., a carefully chosen subset of the data for coreset selection or a synthetically crafted dataset in distillation. However, they often rely on computationally intensive optimization and assume full dataset access during both the selection/distillation and, in many cases, the training process. These added computational costs also affect the overall speed-up possible with these techniques. Finally, these methods tend to be tightly coupled to the model and the task.

Classical lossless data compression can be used to reduce the footprint of the dataset when stored, but require decompression and loading of the uncompressed dataset into memory for the training process, including coreset selection or dataset distillation, if applied. Lossy compression methods are available, but **trade-off storage space for accuracy** and may generate unintended results Underwood et al. (2024). In the case of images, the accuracy-compression trade-off has been shown to be non-linear, dependent on the ML task, model and data Zhao et al. (2020).

This raises a fundamental question: *can we develop a task- and model-independent ML system that can reduce storage costs, provide high memory efficiency during the data selection/distillation process, maintain high accuracy, and result in an overall runtime speed-up during training?*

We propose *dreaMLearning*, a general-purpose framework that enables direct training on compressed data without decompression, streamlining from storage to training Fig. 1 summarizes its benefits based on our experimental results. The framework is built upon Generalized Deduplication (GeDe) Vestergaard et al. (2019), a lossless, random-access compression method. As illustrated in Fig. 2, conventional pipelines typically require decompressing the data followed by (iterative) coreset selection through computationally intensive optimization procedures. In contrast, dreaMLearning directly produces training-ready compressed datasets that retain the

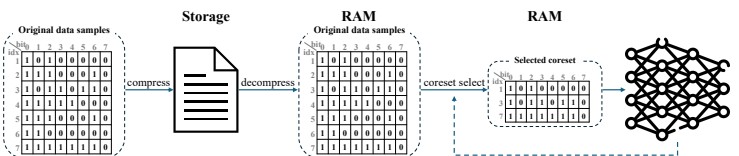

(a) Conventional data-efficient ML pipeline.

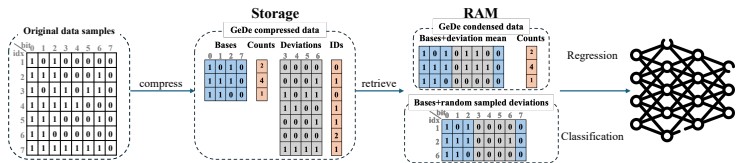

(b) dreaMLearning framework.

Figure 2: A comparison between the existing pipeline and the proposed dreaMLearning framework. dreaMLearning saves storage, memory, and runtime by enabling training directly on compressed data without decompression or subset selection.

essential characteristics required for effective learning. By removing the need for decompression or subset selection, our approach substantially reduces storage requirements, memory footprint, and runtime overhead. Fig.3 demonstrates the effectiveness of dreaMLearning using a simple linear regression example with gradient descent, a fundamental ML algorithm. Fig. 3a shows that, under a uniform per-step cost, training on 45 compressed samples converges faster than full-batch gradient descent on the 1,000-sample dataset. It also more closely tracks the full-batch gradient than mini-batch gradient descent (batch size 45), thereby reducing oscillations and accelerating convergence. As shown in Fig. 3b, the compressed samples capture the underlying distribution, including outliers, preserving characteristics required for robust learning.

This paper shows that the dreaMLearning framework can be implemented with different degrees of complexity and adapt to required data characteristics and performance demands. For example, simpler data sets and problems can rely on a fixed selection of samples, while more complex data sets may introduce lightweight random sampling approaches, and even frequency-domain transformations to improve compressibility for storage in some cases while maintaining memory and accuracy gains. This flexible design allows dreaMLearning to support diverse data modalities, model architectures, and learning tasks.

This work makes the following contributions. First, we introduce dreaMLearning, a unified framework that enables training directly on compressed data, eliminating the need for decompression or subset selection. Second, we propose EntroGeDe, an entropy-based extension of Generalized Deduplication that jointly optimizes information retention and compression efficiency, two traditionally conflicting objectives in prior GeDe methods. Third, through extensive experiments on tabular and image datasets, we show that dreaMLearning significantly improves training speed, memory usage, and storage efficiency, while maintaining competitive accuracy.

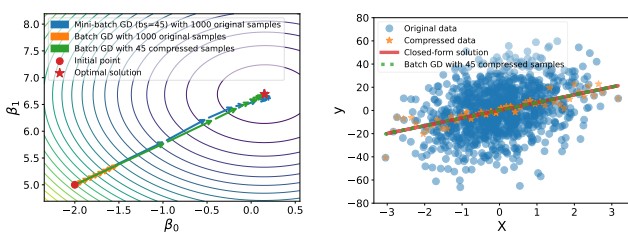

(a) Convergence contour of GD.  (b) Compressed and original data.

Figure 3: Linear regression with gradient descent (GD).

## 2 BACKGROUND AND RELATED WORK

This section reviews foundational techniques that underpin our approach, focusing on two key areas: data compression via GeDe and coreset selection for efficient model training. We first describe GeDe, which enables lossless compression with efficient random access, and discuss recent extensions to various data types and analytical tasks. Next, we survey coreset selection methods, and highlight their limitations in scalability and computational overhead. Finally, we introduce our proposed method, dreaMLearning, which unifies these two techniques to enable efficient model training directly on compressed data.

### 2.1 GENERALIZED DEDUPLICATION

Deduplication Quinlan & Dorward (2002); Meyer & Bolosky (2012) compresses data by replacing identical data chunks with pointers to their first occurrence. GeDe extends this technique to encompass similar, albeit non-identical data chunks. As illustrated in Figure 4, GeDe splits data chunks into frequently appearing parts, *bases*, and high-variance parts, *deviations*, and deduplicates bases and stores deviations unchanged alongside pointers to corresponding bases to enable lossless decompression. Unlike many other lossless compression algorithms, GeDe also supports efficient random access Vestergaard et al. (2020).

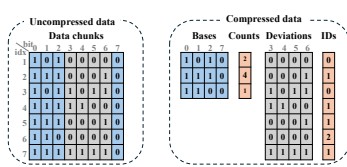

Figure 4: An example of GeDe applied to 8-bit data chunks.

The key question that determines GeDe compression performance is how and where to split data chunks into bases and deviations. In general, allocating more data to the *base* (i.e., the collections of all bases) improves compression since more data is deduplicated, but can also increase the number of unique bases, which reduces compression Hurst et al. (2022). Various heuristic methods have been proposed for allocating data between base and deviation. One early approach pre-computes inter-bit correlations and uses the maximum correlation for each bit to select bits for allocation to the base Vestergaard et al. (2020). A recent variant, GreedyGD Hurst et al. (2024), uses a greedy search algorithm to iteratively minimize the number of new bases created by

enlarging the base. GreedyGD also introduces data pre-processing that significantly improves compression using GeDe; it has been further refined for floating-point data Taurone et al. (2023). GeDe has also been applied to image compression, where it performs particularly well Rask & Lucani (2024).

As it enables efficient random access, GeDe also facilitates analytics directly on compressed data without decompression Hurst et al. (2021). Moreover, accessing only the bases elements suffices for approximate analytics, accelerating tasks. For instance, one may perform highly accurate $k$-means clustering on GeDe-compressed data much faster and with lower memory usage than with uncompressed data Hurst et al. (2021; 2022). Similar results have also been achieved for anomaly detection Taurone et al. (2024), while allocating additional bits to the base improves performance Hurst et al. (2022).

## 2.2 CORESET SELECTION

Coreset selection (CS) identifies subsets of data that retain the essential learning characteristics of the full dataset. Several strategies have been proposed, including geometry-based methods Sener & Savarese (2018); Sinha et al. (2020), uncertainty-based methods Coleman et al., and error/loss-based methods Toneva et al. (2019); Paul et al. (2021). The state-of-the-art rests on gradient-based methods Mirzasoleiman et al. (2020); Killamsetty et al. (2021a;b), which leverage gradients computed during training to select data points aligned with the model's optimization dynamics. CRAIG Mirzasoleiman et al. (2020) and GRAD-MATCH Killamsetty et al. (2021a) iteratively select subsets whose gradients match those of the full data, leading to comparable training dynamics. GLISTER Killamsetty et al. (2021b) formulates subset selection as a bi-level optimization problem, maximizing validation performance rather than minimizing training loss. Still, these approaches are computationally intensive, as they require multiple rounds of gradient computation and optimization, and become particularly burdensome on large-scale datasets or complex models with costly gradient evaluations. InfoBatch Qin et al. (2023) introduces an unbiased dynamic data pruning strategy that accelerates training by adaptively discarding redundant samples without explicit gradient matching. However, its validation is limited to settings where at least 30% of the training data is retained, and its applicability in more aggressive data reduction regimes remains unclear.

A complementary line of work fixes the subset to avoid update overhead. TDDS Zhang et al. (2024) ranks samples based on their contribution to training, combining temporal consistency with gradient-based metrics. Nevertheless, TDDS requires training on the full data upfront. Dataset Quantization (DQ) Zhou et al. (2023) clusters data points and forms a coreset by representatives from each cluster guided by submodular maximization Iyer et al. (2021) to effectively reduce data size while preserving essential information. Nevertheless, a fixed subset may not always optimize model performance.

Contrariwise to these methods, dreaMLearning integrates lossless data compression into coreset construction into a pipeline to enable scalable and effective model training from compact data representations. It employs *entropy-based generalized deduplication* (EntroGeDe) to extract a compact set of aggregate representations from clusters of similar data points, which form *condensed* rather than raw samples, thereby capturing rich statistics. The extracted coreset is *training-ready* without decompression or selection. Selection and compression are *entropy-driven*, offering a trade-off between compression power and learning utility.

## 3 ENTROPY BASED GENERALIZED DEDUPLICATION

In existing GeDe-based compression methods Vestergaard et al. (2020); Hurst et al. (2022; 2024); Taurone et al. (2023); Rask & Lucani (2024); Hurst et al. (2021); Taurone et al. (2024), the base can be accessed directly without decompression. Combined with some (small) meta-data generated during the compression process, e.g., the number of times each base is used, these bases can serve as an interesting summary of the original data that support approximate, yet very accurate analytics, e.g., Hurst et al. (2022; 2024). Selecting base bits for GeDe typically balances two conflicting objectives: compression efficiency and analytics utility.

If the selected bits that form the bases generate a lot of duplicates of the bases for each sample, i.e., few unique bases compared to the number of samples in the dataset, this results in a high compression. However, a given bit selection also affects the quality of meaningful analytics in two ways. First, a larger number of bases results in a richer summary for the data, while potentially reducing compression efficiency. Second, if bits from say a given dimension corresponds to the most significant bits for the value of the data, then the analytic calculation is likely to be more accurate compared to choosing the least significant bits. For analytics calculations, existing methods assume deviation bits to be zero (Hurst et al., 2021; 2022; 2024).

In contrast to past methods, EntroGeDe leverages entropy per bit position to guide both clustering and compression processes, as outlined in Algorithm 1. This entropy-guided selection allows EntroGeDe to balance information retention and data deduplication. Additionally, we compute the average deviation per base to improve accuracy compared to previous zero-filling strategies.

**Intrinsic clustering process** We preprocess the dataset using the GreedyGeDe method (Hurst et al., 2021), converting it to binary format and calculating the entropy of each bit position. For each of the $d$ columns, we select the most significant non-constant bits (MSBs), rank them by decreasing entropy, and choose the top $\beta$ bits, proceeding to the next MSBs if $\beta > d$ until $\beta$ is reached. Using these $\beta$ bits, we cluster the $n$ data points into $m$ clusters ($m \ll n$) based on their unique bit combinations. The centroid of each cluster serves as a condensed sample, weighted by the cluster size. These samples are appended to the original dataset for further compression, with minimal impact on the compression ratio due to $m \ll n$. In storage-constrained environments, we store only the $\beta$ bit positions, generating representative samples on demand.

---

**Algorithm 1** Entropy-based Generalized Deduplication

**Require:** Dataset $\mathcal{D}$ with $n$ data points; analytics bit count $\beta$; plateau threshold $\tau$

**Ensure:** Compressed dataset $\mathcal{D}'$ consisting of selected base bit positions $B^*_{\text{pos}}$, bases $\mathbf{B}$, deviations $\boldsymbol{\Delta}$ with base IDs, and condensed sample weights $\mathbf{w}$

1: Preprocess and convert $\mathcal{D}$ to binary format; let $l_t \leftarrow$ total number of bits per data point
2: Compute entropy $H(p_i)$ for each bit position $p_i$ in $\mathcal{D}$
3: *// Clustering phase (prioritizing high-entropy bits)*
4: Consider MSBs of all $d$ columns first; if $\beta > d$, proceed to next MSBs
5: Sort bit positions in decreasing order of entropy
6: Select top $\beta$ bit positions for clustering
7: Cluster data points based on matching values at selected bit positions
8: Compute cluster centroids as condensed samples; record cluster sizes as weights $\mathbf{w}$
9: *// Compression phase (prioritizing low-entropy bits)*
10: Initialize base bit positions $B_{\text{pos}} \leftarrow$ constant positions ($v$ bits), set number of bases $n_b \leftarrow 1$, base length $l_b \leftarrow v$, deviation length $l_d \leftarrow l_t - l_b$, plateau counter $c \leftarrow 0$
11: Compute initial size $S^*$ using Equation 1
12: Sort remaining bit positions by increasing entropy
13: **for** each position $p_i$ in sorted list **do**
14:     Add $p_i$ to $B_{\text{pos}}$
15:     Update bases $\mathbf{B}$ at $B_{\text{pos}}$
16:     Update $n_b \leftarrow |\mathbf{B}|$, $l_b \leftarrow l_b + 1$, $l_d \leftarrow l_d - 1$
17:     Compute compressed size $S$
18:     **if** $S < S^*$ **then**
19:         $S^* \leftarrow S$, $B^*_{\text{pos}} \leftarrow B_{\text{pos}}$
20:         $c \leftarrow 0$
21:     **else**
22:         $c \leftarrow c + 1$
23:         **if** $c \geq \tau$ or $n_b = n$ **then**
24:             **break**
25:         **end if**
26:     **end if**
27: **end for**
28: **return** $\mathcal{D}'$: $B^*_{\text{pos}}$, $\mathbf{B}$, $\boldsymbol{\Delta}$ with base IDs, and weights $\mathbf{w}$

---

**Compression considerations** To maximize duplicate patterns, we select base bits in order of increasing entropy, prioritizing low-entropy bits shared across data points. We initialize with constant bit positions $B_{\text{pos}}$

of size $v$, setting the number of bases $n_b = 1$, base bit length $l_b = v$, deviation bit length $l_d = l_t - l_b$, where $l_t$ is the total bit count, and plateau counter $c = 0$. Each deviation's base ID uses $\lceil \log_2(n_b) \rceil$ bits, and each condensed sample's weight uses $\lceil \log_2(n) \rceil$ bits. The initial best compressed size $S^*$ is:

$$S = n_b l_b + (n + m)(\lceil \log_2(n_b) \rceil + l_d) + m \lceil \log_2(n) \rceil + S_{\text{params}}, \tag{1}$$

where $n_b l_b$ denotes the size of the bases. $(n + m)(\lceil \log_2(n_b) \rceil + l_d)$ represents the size of deviations, including deviation bits and base IDs. $m \lceil \log_2(n) \rceil$ accounts for the weights' size. $S_{\text{params}}$, the size of compression parameters, is typically negligible. Non-constant bit positions are sorted by increasing entropy. Each position $p_i$ is added to $B_{\text{pos}}$, updating $n_b$ based with the number of unique bases at $B_{\text{pos}}$, incrementing $l_b$ by 1, and decrementing $l_d$ by 1. The compressed size $S$ is recomputed using Equation 1. If $S$ is less than the current best size $S^*$, we update $S^*$ and $B_{\text{pos}}$ and reset $c$ to 0. Otherwise, $c$ is incremented by 1. The process terminates when $c$ reaches the threshold $\tau$ or $n_b = n$, indicating no further compression. The hyperparameter $\tau$ balances compression efficiency and computational cost. Upon completion, the bases, base IDs, and deviations for the optimal compression are stored.

## 4 DIRECT LEARNING ON COMPRESSED DATA

DreaMLearning operates on bit-level data across domains, independently of specific tasks or model architectures. To demonstrate this versatility, we illustrate direct learning on compressed data through with two tasks: regression on tabular data and classification on image data.

### 4.1 REGRESSION WITH TABULAR DATASETS

#### 4.1.1 COMPRESSION

Given a tabular dataset $\{(\mathbf{x}_i, y_i)\}_{i=1}^n$ with features $\mathbf{x}_i \in \mathbb{R}^d$ and targets $y_i \in \mathbb{R}$, we apply the EntroGeDe algorithm (Algorithm 1) for compression. This yields $m$ condensed samples $(\mathbf{x}_j^c, y_j^c)$ with weights $w_j$, where $m \ll n$. Each sample summarizes a cluster of original data points, preserving essential patterns for learning. Given EntroGeDe's fine-grained random access property, these weighted samples can be directly used for model training without full decompression, offering both computational efficiency and data fidelity.

#### 4.1.2 LEARNING

Linear regression estimates a parameter vector $\boldsymbol{\theta} \in \mathbb{R}^d$ to model the relationship between inputs and targets via $\hat{y}_i = \mathbf{x}_i^T \boldsymbol{\theta}$. The mean squared error (MSE) loss is defined as:

$$J(\boldsymbol{\theta}) = \frac{1}{2n} \sum_{i=1}^n \left( \mathbf{x}_i^T \boldsymbol{\theta} - y_i \right)^2. \tag{2}$$

The optimal parameters, $\boldsymbol{\theta}^*$, minimize this loss function, typically achieved through gradient descent (GD), which iteratively updates the parameters in the direction of the negative gradient, as

$$\boldsymbol{\theta}_{t+1} = \boldsymbol{\theta}_t - \frac{\alpha}{n} \sum_{i=1}^n \nabla_{\boldsymbol{\theta}} J(\boldsymbol{\theta}_t) = \boldsymbol{\theta}_t - \frac{\alpha}{n} \sum_{i=1}^n \left( \mathbf{x}_i^T \boldsymbol{\theta}_t - y_i \right) \mathbf{x}_i, \tag{3}$$

where $\alpha$ is the learning rate, and $\nabla_{\boldsymbol{\theta}} J(\boldsymbol{\theta}_t)$ is the gradient of the loss function with respect to the parameters $\boldsymbol{\theta}$ at iteration $t$. To support learning on compressed data, we adapt this process using the $m$ weighted condensed samples from EntroGeDe. The weighted loss becomes:

$$J_c(\boldsymbol{\theta}) = \frac{1}{2n} \sum_{j=1}^m w_j \left( \mathbf{x}_j^{c\,T} \boldsymbol{\theta} - y_j^c \right)^2, \tag{4}$$

with the corresponding update rule:

$$\boldsymbol{\theta}_{t+1} = \boldsymbol{\theta}_t - \frac{\alpha}{n} \sum_{j=1}^{m} w_j \left( \mathbf{x}_j^{cT} \boldsymbol{\theta}_t - y_j^c \right) \mathbf{x}_j^c. \tag{5}$$

This formulation enables efficient training directly on EntroGeDe compressed data, significantly reducing computational overhead while preserving the underlying structure and predictive power of the original data.

### 4.2 CLASSIFICATION WITH COMPRESSED IMAGE DATASETS

Direct learning from compressed images is harder than tabular regression due to high dimensionality and inherent variability. The key issues are: (i) condensed samples may not capture full dataset diversity; (ii) limited inter-sample similarity reduces deduplication/compression efficacy. We address this by (a) randomly sampling images for training and (b) applying a frequency-domain transform to improve compressibility.

#### 4.2.1 COMPRESSION

As shown in Section 5, randomly sampled subsets achieve performance comparable to state-of-the-art coreset selection methods with substantially lower computational cost. To leverage this efficiency, we adopt a simplified EntroGeDe approach that omits clustering and focuses exclusively on compression. Consequently, $m = 0$ in Equation 1, as no condensed samples are generated.

We adopt class-wise compression to reduce computational overhead and exploit intra-class similarities, achieving better compression ratios than cross-class approaches. We apply the Discrete Cosine Transform (DCT) to shift images from the spatial to the frequency domain. This provides two key advantages: (1) concentration of energy in fewer coefficients, enhancing compressibility, and (2) revealing latent similarities not apparent in the spatial domain. The transformation introduces negligible data loss, limited to minor rounding errors with no significant impact on ML/DL performance. Each RGB image is first converted to the YCbCr color space (without subsampling), then DCT is applied independently to each channel. EntroGeDe is then used to compress the transformed data, significantly improving compression efficiency.

#### 4.2.2 LEARNING

During training, a random subset of compressed images is retrieved directly from storage, avoiding full dataset decompression. This low-cost access allows subsets to be updated each epoch, maintaining exposure to diverse data throughout training. For DCT-compressed datasets, each retrieved image is first inverse-transformed to the spatial domain and then converted back to RGB. Training follows standard procedures without the need of modifying the loss function or other components.

### 4.3 ADVANTAGES OF DREAMLEARNING

The dreaMLearning framework, which integrates compression and learning, offers several advantages over traditional methods. First, it reduces storage requirements through effective compression. Second, it minimizes memory usage by loading only necessary image subsets into RAM. Third, it accelerates training due to the smaller dataset size. Fourth, it eliminates the need for full dataset decompression, streamlining the pipeline. These advantages come with minimal impact on model performance, as demonstrated in Section 5.

## 5 EXPERIMENTS

This section evaluates dreaMLearning's performance across multiple datasets and models, benchmarking it against state-of-the-art methods. We assess MSE/accuracy, total training time, peak RAM, and storage.

**Baselines**   For regression, we compare dreaMLearning to full training. For classification, we evaluate against GRAD-MATCH Killamsetty et al. (2021a), GLISTER Killamsetty et al. (2021b), DQ Zhou et al. (2023), TDDS Zhang et al. (2024), InfoBatch Qin et al. (2023), plus full-data and random sampling. Following original protocols, GRAD-MATCH/GLISTER update every 20 epochs; DQ/TDDS use fixed subsets. InfoBatch prunes dynamically under the same training budget. dreaMLearning and random sampling update each epoch.

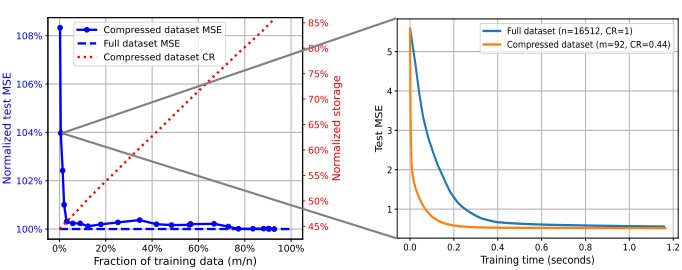

(a) Data fraction vs. MSE & storage.   (b) Training time vs. MSE.

Figure 5: Linear regression on California Housing.

**Datasets and models**   We evaluate on California Housing Pace & Barry (1997), CIFAR-10/100 Krizhevsky et al. (2009), and ImageNet-1K Deng et al. (2009), using linear regression (gradient descent) for California Housing and ResNet-18 for the others. California Housing: 20,640 samples (8 features, 1 target), split 80% train / 20% test. CIFAR-10/100: 50,000 $32 \times 32$ color training images and 10,000 test images; 10 and 100 classes, respectively. ImageNet-1K: 1,281,167 training and 50,000 validation images across 1,000 classes.

Table 1: CIFAR-10 and CIFAR-100 results across subset sizes.

| | CIFAR-10 | | | | | | | CIFAR-100 | | | | | | |
| | 5% | | 10% | | 20% | | | 5% | | 10% | | 20% | | |
| Method | Acc. | Time | Acc. | Time | Acc. | Time | Storage | Acc. | Time | Acc. | Time | Acc. | Time | Storage |
|---|---|---|---|---|---|---|---|---|---|---|---|---|---|---|
| Full | 95.2 | 100% | 95.2 | 100% | 95.2 | 100% | 100% | 78.2 | 100% | 78.2 | 100% | 78.2 | 100% | 100% |
| Random | 88.6 | 5.6% | 90.2 | 10.7% | 93.6 | 20.5% | 100% | 65.2 | 5.6% | **69.9** | 10.9% | 74.2 | 20.4% | 100% |
| GRAD-MATCH | 83.9 | 14.9% | 89.2 | 28.7% | 92.5 | 61.5% | 100% | 46.8 | 14.9% | 61.3 | 21.7% | 68.9 | 40.0% | 100% |
| GLISTER | 75.6 | 10.8% | 88.7 | 19.1% | 92.0 | 35.3% | 100% | 40.3 | 10.8% | 58.3 | 18.8% | 68.9 | 36.2% | 100% |
| DQ | 67.0 | 15.0% | 78.7 | 20.2% | 86.7 | 29.9% | 100% | 20.4 | 15.0% | 36.6 | 25.9% | 55.0 | 37.5% | 100% |
| TDDS | 73.8 | 97.0% | 85.2 | 103.6% | 90.8 | 119.2% | 100% | 32.7 | 97.0% | 51.9 | 106.3% | 62.9 | 117.7% | 100% |
| InfoBatch | 62.5 | 5.7% | 70.6 | **10.4%** | 73.6 | **19.4%** | 100% | 33.5 | **5.1%** | 56.7 | **10.3%** | 59.9 | **20.3%** | 100% |
| **dreaMLearning** | **88.9** | 5.6% | **90.2** | 10.6% | **93.7** | 20.5% | **80%** | **65.4** | 5.6% | 69.6 | 10.9% | **74.2** | 20.6% | **73%** |

**Training settings**   For regression, training on compressed data uses condensed samples with associated weights stored in the compressed representation. We train linear regression with batch gradient descent (learning rate 0.001) until convergence on an Apple M3 Pro (18 GB RAM), averaging results over 10 runs. For classification, CIFAR-10/100 use ResNet-18 for 200 epochs (batch size 128, SGD optimizer, cosine learning-rate decay, initial learning rate 0.05, weight decay $5 \times 10^{-4}$, momentum 0.9). Training runs on an NVIDIA Tesla P100 GPU, with results averaged over 5 runs. ImageNet-1K uses ResNet-18 for 90 epochs (batch size 256, SGD optimizer, step learning rate decays with step size 30 and gamma 0.1, initial learning rate 0.1, weight decay $10^{-4}$, momentum 0.9). Training runs on an NVIDIA GeForce RTX 4090. Due to computational cost, experiments are run once.

**Metrics**   We report test MSE/accuracy, total training time, peak RAM usage of training data, and storage. Except the test MSE/accuracy, all metrics are normalized to the full-data training baseline (set to 1).

**Regression results**   Figure 5a shows the MSE of linear regression on the California Housing dataset across varying fractions of condensed data and their storage requirements. Using condensed data equivalent to 5% of the training set, dreaMLearning achieves performance comparable to the full dataset while requiring less

than 50% of the storage. Figure 5b demonstrates that dreaMLearning, with 92 condensed samples (0.6% of training data, 44% storage), yields an MSE only 4% higher than the full dataset. Its entropy-based clustering in EntroGeDe accelerates convergence by efficiently compressing information into fewer samples.

**Classification results**    Table 1 summarizes CIFAR-10/100 performance across 5%, 10%, and 20% subset budgets. dreaMLearning achieves the best or tied-best accuracy among subset methods while maintaining near-linear scaling in time with the budget. On CIFAR-10, dreaMLearning achieves 88.9% / 90.2% / 93.7% accuracy with 5% / 10% / 20% budgets, respectively, nearly matching full-data performance (95.2%) at just 10%. On CIFAR-100, it attains 65.4% / 69.6% / 74.2% accuracy with 5% / 10% / 20% budgets, outperforming all baselines at each budget except for a marginal difference with Random at 10% (+0.3%). Compared to GRAD-MATCH and GLISTER, dreaMLearning delivers higher accuracy at every budget with much lower time (e.g., 5.6% vs 14.9% at 5% on both datasets). Fixed-subset methods (DQ, TDDS) incur large accuracy losses, and while InfoBatch is slightly faster at some budgets (e.g., 10.4% vs 10.6% time on CIFAR-10 at 10%), it suffers substantial accuracy drops (70.6% vs 90.2% on CIFAR-10; 56.7% vs 69.6% on CIFAR-100 at 10%). dreaMLearning lowers storage to 80% (CIFAR-10) and 73% (CIFAR-100), and constrains peak data RAM to the subset size by streaming and materializing only needed compressed items instead of first loading the entire dataset as baselines do. This eliminates transient full-dataset memory overhead and enables scaling to larger regimes. Overall, dreaMLearning consistently provides the best accuracy-time trade-off across all budgets, while also reducing storage and ensuring memory usage scales proportionally with the subset size.

On ImageNet-1k, prior work identifies InfoBatch as the strongest scalable subset baseline Qin et al. (2023). Accordingly, we benchmark dreaMLearning against InfoBatch, random sampling, and full-data training. Under a 10% subset size, results are summarized in Table 2. dreaMLearning matches the fastest baseline in training time while attaining the highest subset accuracy (59.9% vs. 59.7% for Random and 56.1% for InfoBatch) and reduces storage to 63% of the full dataset. Due to the dataset's large size, data must be streamed

Table 2: ImageNet-1K 10% results.

| Method | Acc. | Time | Storage |
|---|---|---|---|
| Full | 69.1 | 100% | 100% |
| Random | 59.7 | 10.0% | 100% |
| InfoBatch | 56.1 | 10.0% | 100% |
| **dreaMLearning** | **59.9** | **10.0%** | **63%** |

from disk, and peak data RAM is identical across methods and determined solely by batch size and image dimensions. dreaMLearning can further reduce disk I/O by caching most shared bases in memory so that only per-image deviations while training. Future work will quantify these I/O savings and refine cache policies.

## 6    CONCLUSION AND FUTURE WORK

We introduce dreaMLearning, a unified framework that seamlessly integrates entropy-based generalized deduplication (EntroGeDe) with machine learning, enabling efficient training directly on compressed data without decompression. Our approach leverages EntroGeDe to produce condensed, weighted samples for regression tasks on tabular data, while incorporating lightweight random sampling and frequency-domain transformations for classification on high-dimensional image datasets. Extensive evaluations on datasets such as California Housing, CIFAR-10/100, ImageNet-1K demonstrate substantial gains in training speed, memory efficiency, and storage requirements, all while maintaining competitive accuracy. These advancements position dreaMLearning as a promising solution for scalable and efficient learning, particularly in distributed, federated, and edge computing environments. However, the current scope of dreaMLearning is limited to tabular and image data, leaving its performance on other modalities, such as time series, text, or graphs, unexplored. The effectiveness of EntroGeDe hinges data-specific redundancy patterns, which may be compromised by temporal dependencies in time series data or the sparse structures in textual data. To address these constraints, future work will extend dreaMLearning to diverse data modalities, integrate adaptive compression strategies, and further investigate its applications in resource-constrained settings to enhance its robustness and versatility.

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

## APPENDIX

The appendix aims to provide additional details for dreaMLearning focusing on an expanded performance evaluation on the highlighted datasets and problems of the main paper, implications on other key problems (e.g., linear regression, logistic regression) looking at issues of complexity and overall performance, as well as providing a motivation behind the use of the EntroGeDe scheme. Some discussions rely on additional, suitable datasets for the specific problems considered.

## A CLASSIFICATION WITH DIFFERENT DATA FRACTIONS

In the following, we present detailed results derived from experiments conducted on 1%, 5%, 10% and 20% fractions of CIFAR10/100 to highlight the effects of this selection on the accuracy and training time.

Figures 6 and 7 present results on CIFAR-10 and CIFAR-100, respectively. Tables 3 and 4 summarize the details. We analyze dreaMLearning alongside GRAD-MATCH, GLISTER, DQ, TDDS, and InfoBatch. DQ reports no 1% result on CIFAR-100 because its pipeline partitions the training set into 10 bins (5,000 samples per bin, 50 per class), making a 1% per-class selection infeasible. Moreover, DQ consistently underperforms in both accuracy and time relative to the other methods. For example, dreaMLearning reaches $\sim 40\%$ accuracy at 1% of the data, whereas DQ achieves $\sim 20\%$ even at 5%. InfoBatch likewise omits the 1% setting, as this subset size is not supported by its protocol/implementation.

**Accuracy vs. fraction of data** Tables 3 and 4 report accuracy across 1%, 5%, 10%, and 20% budgets for CIFAR-10/100. On CIFAR-10, dreaMLearning attains 71.0%/88.9%/90.2%/93.7% at 1%/5%/10%/20%, respectively, outperforming GRAD-MATCH and GLISTER at all budgets and matching Random at 10% while exceeding it at 5% and 20%. InfoBatch trails substantially at the same budgets (e.g., 70.6% at 10%, 73.6% at 20%). On CIFAR-100, dreaMLearning reaches 38.7%/65.4%/69.6%/74.2%, leading all baselines at 1%, 5%, and 20%; at 10% it is within 0.3 percentage points of Random (69.9%). These results show that dreaMLearning consistently provides the strongest accuracy among subset methods across budgets, especially on the more challenging CIFAR-100.

**Accuracy vs. time** Training time scales near-linearly with budget for dreaMLearning and Random, while subset-selection baselines incur large overheads. On CIFAR-10 at 10%, dreaMLearning trains in 596 s, comparable to Random (601 s) and slightly above InfoBatch (584 s), but with much higher accuracy than InfoBatch (90.2% vs 70.6%). At 5% and 20%, the pattern holds (315/1158 s for dreaMLearning vs 314/1154 s for Random and 324/1094 s for InfoBatch). GRAD-MATCH and GLISTER require substantially more time (e.g., 1617 s and 1076 s at 10%). On CIFAR-100, dreaMLearning remains competitive in time (1%: 89 s; 5%: 313 s; 10%: 597 s; 20%: 1159 s), close to Random (88/312/612/1151 s) and InfoBatch (—/290/581/1146 s), while offering state-of-the-art accuracy among subset methods at most budgets. TDDS consistently exceeds full-data training time even at 10–20% subsets.

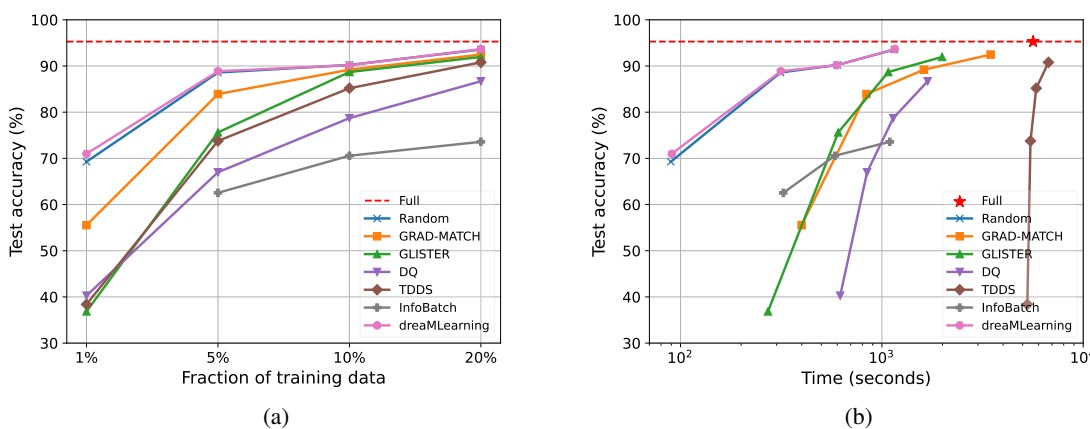

(a)  (b)

Figure 6: Training ResNet18 model on 1%, 5%, 10% and 20% of CIFAR10 dataset. In (a), we show the fraction of data used vs. accuracy. In (b), we show the time vs. accuracy.

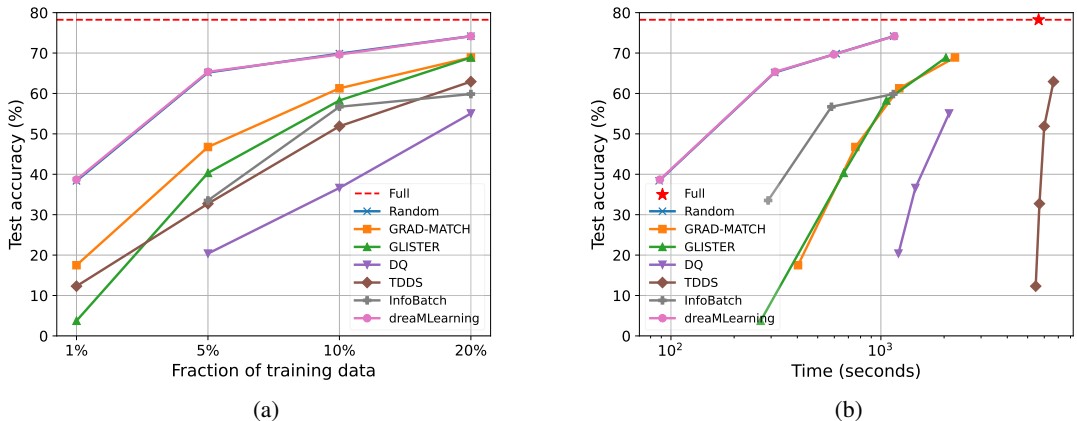

(a)  (b)

Figure 7: Training ResNet18 model on 1%, 5%, 10% and 20% of CIFAR100 dataset. In (a), we show the fraction of data used vs. accuracy. In (b), we show the time vs. accuracy.

## B  THE EFFECTIVENESS OF ENTROPY BASED GEDE

In GeDe-based compression methods, selecting base bits is crucial for performance, as it determines how data is split into bases and deviations for compression. The proposed EntroGeDe method selects base bits based on their entropy, which measures information content at each bit position. The entropy is calculated as:

$$H(X) = -p_i \log_2 p_i - (1 - p_i) \log_2(1 - p_i), \tag{6}$$

where $p_i$ is the probability of a bit being 1 at the $i$-th position. High-entropy bits, with balanced 0s and 1s, indicate high information content and are prioritized for analytics. Conversely, low-entropy bits, with skewed distributions, contain more redundancy and are selected for compression. EntroGeDe leverages high-entropy

Table 3: CIFAR10: Accuracy (%) and Time (seconds) for different data fractions

| Method | 1% | | 5% | | 10% | | 20% | |
|---|---|---|---|---|---|---|---|---|
| | Acc. | Time | Acc. | Time | Acc. | Time | Acc. | Time |
| Full | $95.2 \pm 0.16$ | $5640 \pm 14.2$ | $95.2 \pm 0.16$ | $5640 \pm 14.2$ | $95.2 \pm 0.16$ | $5640 \pm 14.2$ | $95.2 \pm 0.16$ | $5640 \pm 14.2$ |
| Random | $69.3 \pm 1.71$ | $\mathbf{89 \pm 0.4}$ | $88.6 \pm 0.42$ | $\mathbf{314 \pm 0.3}$ | $90.2 \pm 0.22$ | $601 \pm 0.2$ | $93.6 \pm 0.12$ | $1154 \pm 3.0$ |
| GRAD-MATCH | $55.5 \pm 1.52$ | $399 \pm 5.5$ | $83.9 \pm 0.38$ | $838 \pm 6.7$ | $89.2 \pm 0.30$ | $1617 \pm 66.4$ | $92.5 \pm 0.15$ | $3468 \pm 12.2$ |
| GLISTER | $36.9 \pm 2.34$ | $272 \pm 2.6$ | $75.6 \pm 0.77$ | $607 \pm 7.2$ | $88.7 \pm 0.59$ | $1076 \pm 30.6$ | $92.0 \pm 0.10$ | $1991 \pm 93.7$ |
| DQ | $40.3 \pm 1.03$ | $621 \pm 5.6$ | $67.0 \pm 1.63$ | $844 \pm 3.1$ | $78.7 \pm 0.64$ | $1139 \pm 20.5$ | $86.7 \pm 0.58$ | $1688 \pm 4.3$ |
| TDDS | $38.4 \pm 0.91$ | $5275 \pm 52.4$ | $73.8 \pm 0.66$ | $5471 \pm 24.3$ | $85.2 \pm 0.08$ | $5844 \pm 58.5$ | $90.8 \pm 0.23$ | $6720 \pm 178.3$ |
| InfoBatch | N/A | N/A | $62.5 \pm 6.79$ | $324 \pm 0.8$ | $70.6 \pm 5.83$ | $\mathbf{584 \pm 1.6}$ | $73.6 \pm 1.33$ | $\mathbf{1094 \pm 2.1}$ |
| dreaMLearning | $\mathbf{71.0 \pm 0.69}$ | $90 \pm 2.5$ | $\mathbf{88.9 \pm 0.27}$ | $315 \pm 0.7$ | $\mathbf{90.2 \pm 0.34}$ | $596 \pm 4.6$ | $\mathbf{93.7 \pm 0.09}$ | $1158 \pm 2.7$ |

Table 4: CIFAR100: Accuracy (%) and Time (seconds) for different data fractions

| Method | 1% | | 5% | | 10% | | 20% | |
|---|---|---|---|---|---|---|---|---|
| | Acc. | Time | Acc. | Time | Acc. | Time | Acc. | Time |
| Full | $78.2 \pm 0.19$ | $5640 \pm 21.6$ | $78.2 \pm 0.19$ | $5640 \pm 21.6$ | $78.2 \pm 0.19$ | $5640 \pm 21.6$ | $78.2 \pm 0.19$ | $5640 \pm 21.6$ |
| Random | $38.4 \pm 0.62$ | $\mathbf{88 \pm 0.2}$ | $65.2 \pm 0.32$ | $312 \pm 0.7$ | $\mathbf{69.9 \pm 0.26}$ | $612 \pm 0.9$ | $74.2 \pm 0.26$ | $1151 \pm 3.4$ |
| GRAD-MATCH | $17.5 \pm 0.25$ | $403 \pm 3.8$ | $46.8 \pm 0.75$ | $756 \pm 8.2$ | $61.3 \pm 0.33$ | $1224 \pm 14.3$ | $68.9 \pm 0.28$ | $2254 \pm 39.8$ |
| GLISTER | $3.7 \pm 1.35$ | $268 \pm 1.9$ | $40.3 \pm 0.51$ | $665 \pm 27.7$ | $58.3 \pm 0.68$ | $1063 \pm 7.9$ | $68.9 \pm 0.36$ | $2041 \pm 74.1$ |
| DQ | N/A | N/A | $20.4 \pm 0.64$ | $1212 \pm 11.3$ | $36.6 \pm 0.71$ | $1460 \pm 19.9$ | $55.0 \pm 0.46$ | $2113 \pm 4.8$ |
| TDDS | $12.3 \pm 0.33$ | $5462 \pm 28.4$ | $32.7 \pm 0.43$ | $5692 \pm 39.8$ | $51.9 \pm 0.12$ | $5998 \pm 32.4$ | $62.9 \pm 0.22$ | $6640 \pm 19.0$ |
| InfoBatch | N/A | N/A | $33.5 \pm 8.44$ | $\mathbf{290 \pm 0.7}$ | $56.7 \pm 3.82$ | $\mathbf{581 \pm 3.5}$ | $59.9 \pm 1.06$ | $\mathbf{1146 \pm 7.7}$ |
| dreaMLearning | $\mathbf{38.7 \pm 0.43}$ | $89 \pm 0.1$ | $\mathbf{65.4 \pm 0.27}$ | $313 \pm 0.5$ | $69.6 \pm 0.39$ | $597 \pm 2.5$ | $\mathbf{74.2 \pm 0.22}$ | $1159 \pm 0.8$ |

bits to generate the condensed samples, and low-entropy bits for effective compression (deduplication). The approach is relatively simple compared to previous GeDe methods, e.g., GreedyGD, due to its ability to manage much larger dimensions in the data than previous schemes.

We validate effectiveness of EntroGeDe using the California Housing dataset Pace & Barry (1997), comparing high- and low-entropy bit selections on linear regression MSE and storage. Figure 8 shows that high-entropy bit selection significantly reduces MSE, confirming superior information retention. Figure 9 demonstrates that low-entropy bit selection substantially lowers storage needs, validating its compression efficiency. Therefore, EntroGeDe was designed to effectively combine high- and low-entropy bit selection to optimize both information retention and compression, providing an improved accuracy-compression trade-off inherent in existing GeDe methods.

## C COMPLEXITY REDUCTION ANALYSIS OF DREAMLEARNING

Training on compressed data substantially reduces complexity compared to using the full dataset, owing to fewer samples. The complexity reduction of dreaMLearning for linear regression and classification tasks is analyzed below under various considerations.

### C.1 LINEAR REGRESSION TASKS

Consider a dataset of $n$ samples, where each sample is represented as a feature vector $\mathbf{x}_i \in \mathbb{R}^d$ and a target value $y_i \in \mathbb{R}$. Linear regression estimates a parameter vector $\boldsymbol{\theta} \in \mathbb{R}^d$ to model the relationship between inputs and targets via $\hat{y}_i = \mathbf{x}_i^T \boldsymbol{\theta}$. In dreaMLearning, we employ $m(m \ll n)$ compressed samples, denoted $\mathbf{x}_j^c$ and $y_j^c$. These samples are generated by EntroGeDe, with associated weights $w_j$ reflecting the number of

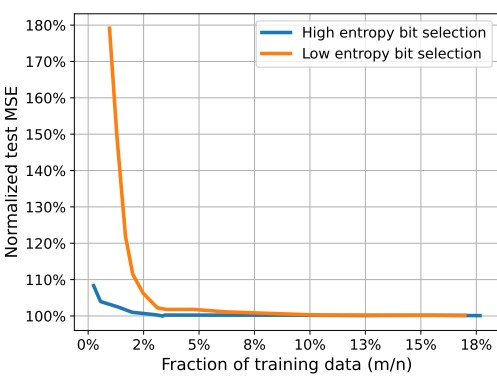

Figure 8: Compressed data fraction vs. MSE.

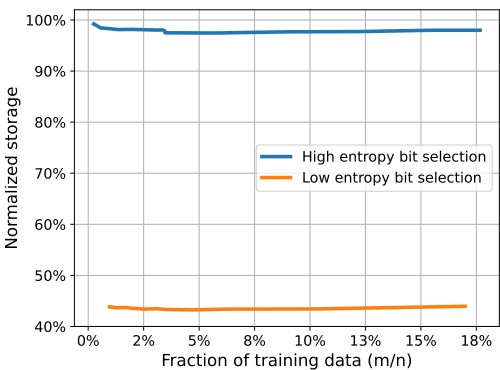

Figure 9: Compressed data fraction vs. storage.

original samples condensed into each compressed sample. The optimal $\boldsymbol{\theta}$ minimizes the error between targets and predicted values, typically using the mean squared error (MSE) loss function, defined as:

$$J(\boldsymbol{\theta}) = \frac{1}{2n} \sum_{i=1}^{n} \left( \mathbf{x}_i^T \boldsymbol{\theta} - y_i \right)^2 . \tag{7}$$

### C.1.1 GRADIENT DESCENT

Gradient descent iteratively updates model parameters to minimize the MSE loss function. The update rule is

$$\boldsymbol{\theta}_{t+1} = \boldsymbol{\theta}_t - \frac{\alpha}{n} \sum_{i=1}^{n} \left( \mathbf{x}_i^T \boldsymbol{\theta}_t - y_i \right) \mathbf{x}_i, \tag{8}$$

where $\boldsymbol{\theta}_t$ is the parameter vector at iteration $t$, $\alpha$ is the learning rate. This incurs $O(nd)$ time complexity per iteration, as gradients are computed for all $n$ samples, yielding $O(knd)$ for $k$ iterations.

In dreaMLearning, the update rule for compressed samples becomes

$$\boldsymbol{\theta}_{t+1}^c = \boldsymbol{\theta}_t^c - \frac{\alpha}{n} \sum_{j=1}^{m} w_j \left( \mathbf{x}_j^{cT} \boldsymbol{\theta}_t - y_j^c \right) \mathbf{x}_j^c. \tag{9}$$

This requires $O(md)$ time complexity per iteration, as gradients are computed for only $m$ samples. Although the weight $w_j$ introduce minor overhead, the computational structure remains unchanged. Thus, dreaMLearning reduces time complexity from $O(knd)$ to $O(kmd)$, a significant improvement when $m \ll n$. Despite potentially more iterations, the smaller sample size substantially lowers overall time complexity.

### C.1.2 OPTIMAL SOLUTION

Alternatively, linear regression has a closed form solution, which directly computes optimal parameters as

$$
\begin{aligned}
\nabla_{\boldsymbol{\theta}} J(\boldsymbol{\theta}) &= \mathbf{0} \\
\Rightarrow \frac{1}{2n} \nabla_{\boldsymbol{\theta}} \left\| \mathbf{X}\boldsymbol{\theta} - \mathbf{y} \right\|_2^2 &= \mathbf{0} \\
\Rightarrow \mathbf{X}^{\mathrm{T}} \mathbf{X} \boldsymbol{\theta} - \mathbf{X}^{\mathrm{T}} \mathbf{y} &= \mathbf{0} \\
\Rightarrow \boldsymbol{\theta}^* &= \left( \mathbf{X}^{\mathrm{T}} \mathbf{X} \right)^{-1} \mathbf{X}^{\mathrm{T}} \mathbf{y}.
\end{aligned}
\tag{10}
$$

This exact solution requires no hyperparameter tuning but is computationally intensive for large or high-dimensional datasets, where we consider $n$ to be the number of samples, and $d$ the dimensionality of the dataset. The time complexity is driven by the matrix multiplication step $\mathbf{X}^{\mathrm{T}}\mathbf{X}$, which is $O(nd^2)$, and inversion step $\left(\mathbf{X}^{\mathrm{T}}\mathbf{X}\right)^{-1}$, which is $O(d^3)$. The time complexity is then $O(nd^2 + d^3)$. Using dreaMLearning with $m \ll n$ compressed samples, the problem can be reduced to an approximate calculation of the form

$$
\begin{aligned}
\nabla_{\boldsymbol{\theta}} J_c(\boldsymbol{\theta}) &= \mathbf{0} \\
\Rightarrow \frac{1}{2n} \nabla_{\boldsymbol{\theta}} \left\| \mathbf{w}^{\frac{1}{2}} (\mathbf{X}_c \boldsymbol{\theta} - \mathbf{y}_c) \right\|_2^2 &= \mathbf{0} \\
\Rightarrow \mathbf{X}_c^{\mathrm{T}} \mathbf{w} \mathbf{X}_c \boldsymbol{\theta} - \mathbf{X}_c^{\mathrm{T}} \mathbf{w} \mathbf{y}_c &= \mathbf{0} \\
\Rightarrow \boldsymbol{\theta}_c^* &= \left( \mathbf{X}_c^{\mathrm{T}} \mathbf{w} \mathbf{X}_c \right)^{-1} \mathbf{X}_c^{\mathrm{T}} \mathbf{w} \mathbf{y}_c,
\end{aligned}
\tag{11}
$$

where $\mathbf{w}$ is a diagonal matrix with weights $w_j$. The additional complexity from weights is negligible as $\mathbf{w}$ is a $m \times m$ matrix. The time complexity reduces to $O(md^2 + d^3)$, significantly lower when $m \ll n$. This reduction enables dreaMLearning to make the calculation feasible for larger datasets, substantially saving time and computational resources. The trade-off incurred is accuracy, as $\boldsymbol{\theta}_c^* \approx \boldsymbol{\theta}^*$.

Consider an example of California housing dataset shown in Figure 5a. The original training set has $n = 16,512$ samples and $d = 8$ features. Using dreaMLearning, we can reduce the dataset to $m = 92$ samples, which is only 0.6% of the original dataset. The time complexity reduction by dreaMLearning is approximately $165\times$ for the optimal solution. By gradient descent, the MSE from dreaMLearning is 4% higher than that of the original dataset, and a similar difference is reasonably expected for the optimal solution.

### C.2 CLASSIFICATION TASKS

For classification tasks, computational complexity varies with model architecture. Therefore, we consider the per-sample complexity of a model, denoted as $O(C)$, which is the cost of a forward and backward pass. The per-epoch complexity is $O(nC)$, where $n$ is the number of training samples. Using compressed data with $m$ samples ($m \ll n$), the per-epoch complexity reduces to $O(mC)$. Consequently, total training complexity decreases from $O(EnC)$ to $O(E'mC)$, where $E$ and $E'$ are the epochs needed for convergence on the original and compressed datasets, respectively. Typically, $E' \approx E$, reducing the computational cost by approximately $n/m$. This significant reduction enhances training speed and lowers memory usage, making dreaMLearning highly efficient for large-scale classification with deep neural networks, as evidenced by our experimental results with ResNet18.

## D LOGISTIC REGRESSION

We evaluate dreaMLearning for logistic regression on the Default of Credit Card Yeh & Lien (2009) and IJCNN1 datasets Chang & Lin (2011). The Default of Credit Card dataset comprises 30,000 samples with 23

features, split 80% for training and 20% for testing. The IJCNN1 dataset includes 49,990 training and 91,701 test samples, each with 22 features. Similar to linear regression tasks, we apply EntroGeDe to compress $n$ training data points to $m$ condensed ones with weights $w$. The logistic regression model is trained using weighted gradient descent on these samples. Figure 10 presents the results for both datasets considering the accuracy achieved (higher is better) and compression rate (lower is better). The test accuracy and storage requirement of compressed data are normalized relative to those of the full dataset. dreaMLearning achieves performance comparable to full-dataset training with significantly fewer samples. For the Default of Credit Card dataset, using 33% of the data yields a 9% accuracy loss with a 75% storage footprint reduction (i.e., compression rate is roughly 0.25), while 55% of the data achieves equivalent accuracy with a 70% reduction. For IJCNN1, 1% of the data attains 90% of full-training accuracy with an 80% storage reduction, and 50% of the data matches full-training accuracy with a 70% reduction. These results demonstrated that dreaMLearning enables substantial data and storage reduction for logistic regression, while maintaining accuracy close to that of full-data training.

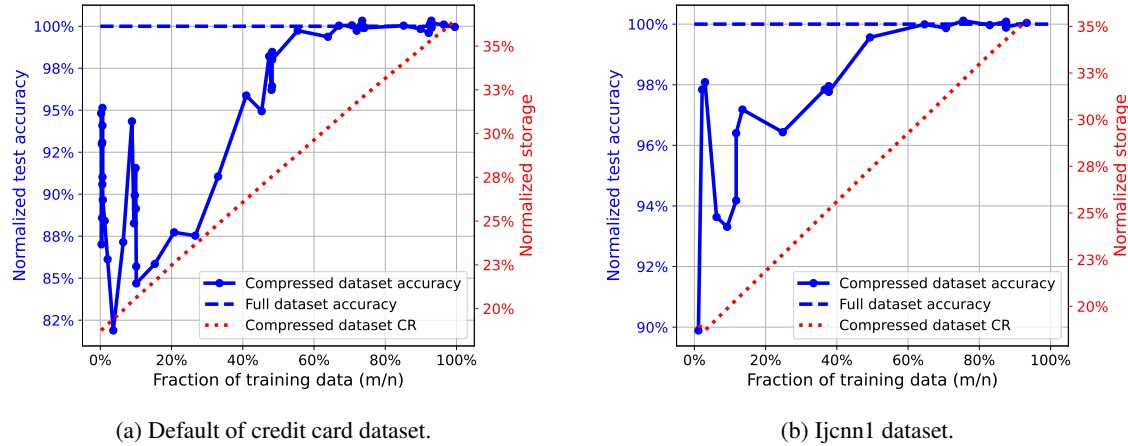

(a) Default of credit card dataset.

(b) Ijcnn1 dataset.

Figure 10: Comparison of full and compressed data for logistic regression tasks.