# OpenReview forum: "dreaMLearning: Data Compression Assisted Machine Learning"
_ICLR.cc/2026/Conference — ICLR 2026 Conference Withdrawn Submission_

### Official Review · Reviewer_3Z4E · 2025-10-28

**Soundness:** 2
**Presentation:** 3
**Contribution:** 2
**Rating:** 4
**Confidence:** 3

**Summary:**

The paper proposes dreaMLearning, a general framework for training machine learning models directly on compressed data without decompression. Built on Entropy-based Generalized Deduplication (EntroGeDe). Compared to conventional pipelines that decompress data and often perform computationally heavy coreset selection or dataset distillation, dreaMLearning produces training-ready compressed datasets and can incorporate lightweight sampling or frequency-domain transforms. Experiments on regression and classification with tabular and image data show some numerical performance of dreaMLearning compared to coreset baselines.

**Strengths:**

The paper proposes dreaMLearning  to train directly on losslessly compressed data, integrating an entropy-guided extension of Generalized Deduplication (EntroGeDe); this creative combination distinguishes it from coreset selection and dataset distillation. Its quality is supported by a principled foundation in GeDe, intuitive mechanisms for information preservation, and empirical evidence across tabular and image tasks showing substantial runtime, memory, and storage gains with minimal accuracy loss, complemented by a linear regression convergence example. Presentation is accessible, with motivation and illustrative figures that effectively contrast conventional pipelines with the proposed framework. Addressing pressing storage and memory bottlenecks for large-scale, distributed, and edge/federated ML, the reported improvements—if broadly validated—could meaningfully influence how datasets are prepared and consumed in practice.

**Weaknesses:**

-The authors claim that training on compressed representations preserves the training-relevant distribution while accelerating convergence. However, there is limited formal analysis connecting EntroGeDe’s compression objective (entropy and deduplication) to the generalization or optimization dynamic.

-Provide theoretical guarantees or bounds (e.g., stability, gradient bias, variance changes…relative to full-batch and mini-batch SGD) for some models beyond linear regression, and include conditions under which the compression preserves key statistics (like class distributions in the case of classification).

-The authors reported speedups and memory/storage reductions focused only on training, but the end-to-end benefits depend on compression time, preprocessing, and random access costs...

-Include wall-clock ablations that separate compression and training time; measure throughput under realistic storage systems.

-The paper spans only tabular and image classification/regression, but modern training often involves diverse architectures and data types. It will be nice to discuss how this can be generalized for these other types of  architectures and data types.

-Expand evaluation to include at least some modern architectures like Transformers, large CNNs…

-How dreaMLearning react in the case of noisy data …

- What is the sensitivity of dreaMLearning to compression parameters? ablations on EntroGeDe hyperparameters missing...

**Questions:**

-The authors claim that training on compressed representations preserves the training-relevant distribution while accelerating convergence. However, there is limited formal analysis connecting EntroGeDe’s compression objective (entropy and deduplication) to the generalization or optimization dynamic.

-Provide theoretical guarantees or bounds (e.g., stability, gradient bias, variance changes…relative to full-batch and mini-batch SGD) for some models beyond linear regression, and include conditions under which the compression preserves key statistics (like class distributions in the case of classification).

-The authors reported speedups and memory/storage reductions focused only on training, but the end-to-end benefits depend on compression time, preprocessing, and random access costs...

-Include wall-clock ablations that separate compression and training time; measure throughput under realistic storage systems.

-The paper spans only tabular and image classification/regression, but modern training often involves diverse architectures and data types. It will be nice to discuss how this can be generalized for these other types of architectures and data types.

-Expand evaluation to include at least some modern architectures like Transformers, large CNNs…

-How dreaMLearning react in the case of noisy data …

- What is the sensitivity of dreaMLearning to compression parameters? ablations on EntroGeDe hyperparameters missing...

---

### Official Review · Reviewer_R211 · 2025-10-31

**Soundness:** 2
**Presentation:** 3
**Contribution:** 2
**Rating:** 4
**Confidence:** 3

**Summary:**

This paper proposes dreaMLearning, a training framework that integrates lossless, random-access compression with learning so that models can train directly on compressed data without explicit decompression as traditional methods did. The method builds on previous work of Generalized Deduplication (GeDe) and introduces Entropy-based GeDe (EntroGeDe), which uses bit-wise entropy to guide both clustering for condensed samples and the allocation of bits to the base for compression efficiency. For tabular regression, the system trains on a small set of condensed, weighted samples produced by EntroGeDe. For image classification, the pipeline applies color-space conversion and DCT to improve compressibility, performs class-wise compression, and refreshes a compressed subset each epoch. Experiments report accuracy, running time, peak data RAM, and storage on California Housing, CIFAR-10/100, and ImageNet-1K. The results show that the proposed EntroGeDe achieves near-full accuracy with 92 condensed samples (about 0.6 percent of data) and 44 percent storage on California Housing dataset, 90.2% at 10 percent subset on CIFAR-10 dataset, and  yields 59.9 percent accuracy with 63 percent storage on ImageNet-1K at 10 percent subset.

**Strengths:**

The strengths can be summarized as follows.
- This paper proposes a unified pipeline that avoids explicit decompression and reports accuracy, time, RAM, and storage with consistent protocols.
- The proposed Entropy-guided EntroGeDe provides a way to trade information retention against compression, with an explicit compressed-size objective and an implementable algorithm.
- The implementation details are clear for training setups and baselines, which improves reproducibility.

**Weaknesses:**

The weaknesses are summarized as listed below.
- **The practical gains can be conservative in large-scale vision.** On ImageNet-1K at 10 percent subset, accuracy is 59.9% vs random 59.7% with identical training time, while storage remains 63% of full. The gain over random is small, and the storage figure limits the headline benefit in this regime.
- **The results of subset training may be confounding.** CIFAR-10 at 10% already reaches 90.2 percent versus 95.2 percent full, which is a well-known behavior of subset training. The paper would be stronger if it disentangled benefits attributable to EntroGeDe versus those due to the epoch-wise subset refresh and standard regularization.
- **System claims under-measured.** The narrative suggests potential I/O and cache wins from reusing bases in memory, but these gains are not quantified. The paper explicitly lists cache and I/O evaluation as future work.
- **Vision pipeline relies on conventional transforms.** The image route leans on YCbCr conversion and DCT before GeDe, then uses per-epoch random subset selection. The originality on the vision side feels closer to an engineering recipe that stitches known transforms with the proposed compressed training loop.

**Questions:**

Q1. Can you ablate the vision pipeline to isolate contributions of color-space conversion, DCT, class-wise compression, and epoch-refresh sampling? Please report accuracy, time, RAM, and storage for each removal.

Q2. How does EntroGeDe compare to GreedyGD on the exact same datasets when both are used for compressed training, not just for compression quality? A head-to-head would isolate the benefit of entropy-guided allocation.

Q3. On ImageNet 1K with a 10 percent subset the accuracy gain over random is very small while storage remains about 63 percent of full. Can you provide equal-accuracy Pareto curves that show a clear storage reduction, or explain the regimes where your method delivers strong practical wins?

Q4. How portable is the recipe beyond ResNet on CIFAR and ImageNet? Please either show a small example on ViT or another modality, or discuss failure modes. Also provide a brief integration guide for PyTorch users and a comparison against standard pipelines such as JPEG with WebDataset in accuracy, training time, and storage.

---

### Official Review · Reviewer_yMTe · 2025-11-03

**Soundness:** 2
**Presentation:** 2
**Contribution:** 2
**Rating:** 2
**Confidence:** 4

**Summary:**

The paper introduces dreaMLearning, a framework for direct machine learning on compressed data—avoiding full decompression. It builds on Entropy-based Generalized Deduplication (EntroGeDe), which combines clustering on high-entropy bits and compression on low-entropy bits to retain informative patterns while minimizing redundancy. Experiments on regression (California Housing) and classification (CIFAR-10/100, ImageNet-1k) show reduced memory and storage usage with comparable accuracy to standard training.

**Strengths:**

1. Using of bit-level entropy for adaptive clustering and compression improves both information retention and compression efficiency.

2. Evaluations span multiple domains, not only image datasets, but tabular regression.

3. Achieves 10× faster training, 10× lower memory, and ~40% storage savings with minimal accuracy drop; especially promising for edge/federated applications.

**Weaknesses:**

1. I have a big concern about the baseline results reported in the paper. All that seems to be lower than the naive baseline random selection. And all the approaches the shapes lower performance than full data training. For instance, the reported results for InfoBatch are significantly weaker than those in the original paper. At first glance, I thought it was because of the large pruning ratio used in this paper. However, I checked the original infobatch paper and found that they can achieve 94.7% acc on cf10 with a 70% pruning ratio. In this paper, infobatch achieves 73.6% with an 80% pruning rate, which I think is quite low for a simple dataset. As there is no code submitted, I couldn't check if the baselines in this paper are correctly implemented or not. A reproduction check or ablation would be important to ensure fairness.

2. Although the authors emphasize “no decompression,” the image experiments still apply inverse DCT before feeding data to the model, which partially contradicts the stated claim of direct training on compressed data.

3. Key hyperparameters are fixed without justification. The robustness of the approach to these parameters is not demonstrated.

**Questions:**

1. Why does InfoBatch perform so poorly relative to random selection? Were all baselines re-tuned under the same compute and subset constraints?

2. How stable are the entropy-based clusters — do different random seeds lead to different compressed representations?

3. Could the proposed method integrate with self-supervised or large-scale transformer pretraining pipelines?

**Details Of Ethics Concerns:**

It seems that the baseline results are not reliable.

---

### Official Review · Reviewer_zv7b · 2025-11-12

**Soundness:** 2
**Presentation:** 2
**Contribution:** 2
**Rating:** 4
**Confidence:** 4

**Summary:**

This paper introduces dreaMLearning, a novel framework that enables learning from compressed data without decompression, built upon an entropy-driven lossless compression method that consolidates information into a compact set of representative samples.

**Strengths:**

1. A novel method that learns from compressed data without having to decompress, speeding up performance and lowering memory demand, outperforms all related concepts.
o	“accelerates training by up to 10×, reduces peak memory usage of training data by 10×, and cuts storage by 37%, with a minimal impact on model performance.”
2. Per epoch costs decrease significantly.
o	"dreaMLearning delivers higher accuracy at every budget with much lower time"
o	"Training time scales near-linearly with budget for dreaMLearning and Random, while subset-selection baselines incur large overheads"
3. The authors claim the method is flexible across various datasets, and the framework is generalizable.
o	"dreaMLearning accommodates a wide range of data types, tasks, and model architectures"
4.	The case study on California Housing appears strong, showcasing improved training time.
5.	Clear/Clean Algorithm Demonstration

**Weaknesses:**

1. Lacks any strong theoretical section, the method seems to be based on heuristics as opposed to strong mathematics.
o	It includes a few mathematical concepts; they have a bit of mathematics covering things like implementation and how their algorithm can be derived from MSE, but they lack pure proofs.
2. For DCT-compressed images, they inverse-transform and reconvert before training; that’s still a decompression step, which goes against their phrasing
o	"For DCT-compressed datasets, each retrieved image is first inverse-transformed to the spatial domain and then converted back to RGB."
3. Entropy choice spread out; lacks firm directionality.
o The clustering phase prioritizes high-entropy bits
o The compression phase prioritizes low-entropy bits
o "To maximize duplicate patterns, we select base bits in order of increasing entropy, prioritizing low-entropy bits shared across data points."
4. CIFAR-10 does not drop as low as the headline suggests, indicating that the datasets.
o	"dreaMLearning lowers storage to 80% (CIFAR-10) and 73% (CIFAR-100)"
5. Vision results use ResNet-18 only; broader coverage would back generalizability claims.
•	(Minimal but additional) - dreaMLearning is promoted as one of the fastest methods, but gets beaten a few times (however, it still performs very well in terms of time).
6. "InfoBatch is slightly faster at some budgets (e.g., 10.4% vs 10.6% time on CIFAR-10 at 10%)"
•	(Minimal but additional) - Requires class compression, does not compress completely unsupervised, insignificant detail; however, just a limitation.
7. "We adopt class-wise compression to reduce computational overhead and exploit intra-class similarities, achieving better compression ratios than cross-class approaches"

Overall, I have more cons than pros written, but I do think that this paper is still very good. The benefits are substantial enough to outweigh the cons/limitations.Cons:

**Questions:**

See above

---

### Note · Authors · 2025-11-27

I have read and agree with the venue's withdrawal policy on behalf of myself and my co-authors.